# Electron Beam Irradiation Isolates Cellulose Nanofiber from Korea “Tall Goldenrod” Invasive Alien Plant Pulp

**DOI:** 10.3390/nano9101358

**Published:** 2019-09-22

**Authors:** Hong Gun Kim, U Sang Lee, Lee Ku Kwac, Sang Ok Lee, Yong-Sun Kim, Hye Kyoung Shin

**Affiliations:** Institute of Carbon Technology, Jeonju University, 303 Cheonjam-ro, Wansan-gu, Jeonju-si, Jeollabuk-do 55069, Korea; hgkim@jj.ac.kr (H.G.K.); 2dbtkd@naver.com (U.S.L.); kwack29@jj.ac.kr (L.K.K.); lso0594@naver.com (S.O.L.); wva223g6@naver.com (Y.-S.K.)

**Keywords:** Korean invasive alien plants, electron beam irradiation, cellulose nanofibers, tall goldenrod

## Abstract

This work investigates the possibility of isolating cellulose nanofibers from pulps of tall goldenrod plant, which are invasive plants in Korea, by a convenient method, without strong acids or high-pressure homogenization, using electron beam irradiation (EBI). The obtained cellulose nanofibers were characterized by scanning electron microscopy (SEM), ultraviolet–visible (UV–vis) spectroscopy, X-ray diffraction (XRD), thermogravimetric analysis (TGA), and in terms of their mechanical properties. SEM showed that the initially isolated 10-μm-diameter cellulose fibers became more finely separated with increasing EBI dose, and that cellulose fibers treated with 300 kGy of EBI were separated into long cellulose nanofibers of around 160 nm in diameter. In addition, the paper samples prepared from more finely separated fibers generated by using higher doses of EBI had enhanced UV–vis transmittance. Via the XRD analysis, we observed that cellulose I in the EBI-treated cellulose fibers were gradually converted into a different type of cellulose similar to cellulose type II, as the EBI dose increased. Meanwhile, the TGA demonstrated that the finely separated cellulose fibers observed after administering the high EBI dose had lowered thermal stability due to the reduction of cellulose I but higher char yield. In addition, tensile strengths of paper samples increased with decreasing the diameters of their constituent fibers that result from the different EBI doses used in the preparation of the paper pulp.

## 1. Introduction

Invasive alien plants present serious problems for biodiversity. Moreover, alien plants can generate environmental and agricultural threats in the areas in which they invade, and then negatively impact upon natural organization, resulting in enormous economic losses [1,2,3,4,5,6]. Therefore, it is desirable that invasive alien plants are converted into useful materials that can provide environmental and economic benefits.

Tall goldenrod (*Solidago altissima* L.; see Figure 1) is an invasive alien plant in Korea. It is a type of non-wood perennial plant with a height of 1–2.5 m and is a naturalized plant of North American origin [7,8]. Tall goldenrod is mainly distributed in southern areas of Korea and grows in vacant grounds or on roadsides. Similar to wood, non-wood plants such as tall goldenrods are typically composed of cellulose, lignin, and hemicellulose, as well as a small quantity of pectin and pigments. The structure is similar to composites in that celluloses of stiff fillers are surrounded by lignin and hemi-cellulose, which acted as cements [9,10]. Cellulose of semi-crystalline is the main component of such non-wood and wood structures, and provides outstanding mechanical and thermal properties due to strong hydrogen bonding between hydroxyl groups in the linear polymer of *β*-(1→4)-d-glucopyranose units [11,12], whereas the lignin and hemi-cellulose components are amorphous [13]. Thus, cellulose is applied to plastic substitute materials or various reinforcement composites as organic fillers [14,15,16]. In recent years, many researchers have paid close attention to cellulose nanofibers that have been further fibrillated through cleavage of the amorphous parts of micro-sized cellulose by acid hydrolysis [17,18,19,20] or a mechanical process [21,22,23]. Cellulose nanofibers have advantages over micro-sized cellulose fibers, such a high crystallinity, ultralight weight, high stiffness, and high strength, and can therefore be applied to automobiles and aircrafts [24,25,26].

However, cellulose nanofibers are difficult to extract from pulps. Most cellulose nanofibers are extracted via chemical methods such as strong acid hydrolysis, or by mechanical methods, e.g., high pressure homogenization or grinding. However, there are some drawbacks to these extraction process. In the case of strong acid hydrolysis, environmental pollution and low yield owing to the severe reaction conditions are unfortunate side effects. In the case of mechanical extraction, high energy consumption often occurs as a result of the blocking of the homogenizer blade. 

In the present study we use electron beam irradiation (EBI), which can advance the conversion of materials by inducing cross-linking, polymerization [27,28,29,30] and degradation [31] without employing chemical reagents. Thus, EBI may be an environmentally friendly method. Kim et al. [32] prepared nanocellulose using sulfuric acid of 63 wt.% concentration, after EBI treatment, from bleached kenaf core pulp. Lee et al. [33] prepared cellulose nanocrystals using high pressure homogenization treatment and alkaline treatment or/and further oxidation/cationization, after EBI treatment (~3000 kGy), from cellulose pulp. However, Shin et al. [31] studied cellulose fiber isolation using only water, without chemical treatment, in their cooking process, from kenaf bast fibers pre-treated by EBI. Therefore, adopting the approach from the study of Shin et al. [31], here we investigated the isolation of cellulose nanofibers by a convenient method, applying them to the utilization of the invasive alien plant species tall goldenrod. This process avoids the use of strong acid or high-pressure homogenization, using only EBI treatment to eliminate environmental damage via acid contamination or energy consumption. The obtained cellulose nanofibers are characterized using scanning electron microscopy (SEM), UV–vis transmission spectra, X-ray diffraction (XRD), and thermogravimetric analysis (TGA), as well as in terms of their tensile strength.

## 2. Experimental

### 2.1. Materials

Tall goldenrod plants growing on the roadsides of the southern area of Korea in September were directly collected. Tall goldenrod pulp was obtained via alkali cooking and 2 steps of bleaching treatment as detailed in Table 1. The bleached pulps were lyophilized after washed over 10 times in water and then kept in a desiccator. This material was subsequently used as the experimental materials for the preparation of cellulose nanofibers [31]. All chemicals were analytical grade and used as received.

### 2.2. Preparation of Cellulose Nanofibers

To prepare the cellulose nanofibers, cellulose fibers were irradiated with an electron beam. EBI was performed using a scanned beam of 1.14 MeV accelerating voltage, 7.6 mA beam current, 110 cm irradiation width, and with a distance 20 cm between the window and samples; the dose rate was 6.67 kGy/sec at room temperature in an air atmosphere. The absorbed doses that were selected for testing were 50, 100, 200, and 300 kGy. EBI-treated cellulose fibers (0.2 g) were wetted with distilled water (2 g) and manually ground using a mortar and pestle for 1 min. The samples obtained were labeled P-0 (raw pulps), P-50, P-100, P-200, and P-300, for the samples of cellulose fibers treated with EBI doses of 50, 100, 200, and 300 kGy, respectively. Figure 2 schematically shows the process of obtaining cellulose nanofibers using EBI.

### 2.3. Analysis

SEM was used to observe morphology to understand the influence of EBI on the degree of separation of tall goldenrod pulp samples. All samples were coated with gold using a vacuum sputter and then imaged using Jeol JSM 5910 LV (Tokyo, Japan) at 5000× magnification, and HITACHI SU-70 microscopes (Tokyo, Japan), at 10,000× and 50,000× magnification.

Ultraviolet–visible (UV–vis) light transmittance was performed on paper samples (thickness: 0.14 ± 0.023 mm) that had been prepared using only thermocompression method without the use of a binder or adhesive agent, from each of the P-0, P-50, P-100, P-200, and P-300 and was measured at a wavelength in the range of 300 to 800 nm using a U-4100 spectrophotometer (Hitachi High Technologies America Inc., Tokyo, Japan).

X-ray diffraction (XRD) pattern was obtained with using a RIGAKU, D/MAX-2500 instrument (Tokyo, Japan), using Cu Kα radiation at 40 kV and 30 mA, and scanned in the 2*θ* range from 5° to 30° at a scan rate of 0.4°/min.

The thermal stability of the samples was assessed with a thermo-gravimetric analyzer (TA, SDT Q600, New Castle, DE, USA). A sample (10 mg) was placed in an alumina pan and heated over the temperature range from 25 to 950 °C, at a heating rate of 10 °C/min, under nitrogen atmosphere.

Tensile testing was performed with paper samples mentioned in the above UV–vis light transmittance measurement. The measurements were performed using an Instron 5050 tester (Instron USA, Norwood, MA, USA) with a load cell of 1 kN; test samples were cut to a width of 15 mm and a length of 50 mm.

## 3. Results

### 3.1. SEM Images

The SEM images in Figure 3 illustrate the differing fiber separations among the differently dosed EBI-treated tall goldenrod pulp samples. This effect can be explained by the fact that the EBI treatment permeates weak parts between cellulose nanofibers within micro-sized cellulose fibers, inducing wakening and breaking. Then, by manually grinding the cellulose fibers using a mortar and pestle, the degree of separation of within the cellulose fibers is enhanced as the sections between cellulose nanofibers already weakened by EBI break. The higher EBI dose generated more finely cellulose fibers. As shown in Figure 3a, non-EBI-treated pulps have cellulose fibers of larger than or approximately equal to 10 μm in diameter, with smooth, flat, and yet relatively non-uniform exteriors. However, after EBI treatment, rough exteriors with cracks on the fiber surfaces were observed (P-50 and P-100, Figure 3b,c). When the EBI dose is increased to 200 and 300 kGy, tall golden-rod pulp samples of approximately 10 μm in width were separated into long cellulose nanofibers that had diameters of ~800 nm. In particular, in the case of P-300, we observed that the approximately 10-μm pulp sample was separated long cellulose nanofibers of less than 160 nm in diameter, as shown in the high magnification image in Figure 3e. These results indicate that EBI treatment is capable of separating cellulose microfibers within pulp samples into cellulose nanofibers, without any requirement for strong-acid treatment or high-pressure homogenization. Figure 4a exhibits the visible light transmittance for the paper samples prepared from the P-0, P-50, P-100, P-200, and P-300 paper samples with cellulose fibers of different diameters at wavelengths in the range of 300–800 nm. The paper sample prepared from P-0, having fibers of over 10 μm in diameter, is optically opaque and shows approximately 0% light transmittance at wavelengths of 300–800 nm. The paper samples prepared from the P-50 and P-100 papers are also opaque. However, the paper sample prepared from T-200, having fiber diameters of ~800 nm, shows a maximum transmittance of 30%, and in the paper sample prepared from T-300 the transmittance increases to 42% at the wavelength of 500 nm. Figure 4b shows photographs that illustrate the opacity differences among the paper samples with cellulose fibers of different diameters prepared using tall goldenrod pulp obtained after the various EBI doses. The paper samples prepared from P-0, P-50, and P-100, mostly consisting of micro-cellulose fibers, are not transparent, whereas the paper samples prepared from P-200 and P-300, having cellulose nanofibers, are more transparent. These results allow us to conclude that in the paper samples produced from the tall goldenrod pulps, the finer separation of fibers with increasing EBI dose to the pulp contributes to higher transmittance.

### 3.2. XRD Analysis

The XRD spectra plotted in Figure 5 show the crystalline structure variation between the P-0 and P-50, P-100, P-200, and P-300 pulp samples, obtained by treating tall goldenrod pulp with various EBI dose. As shown in Figure 5, non-EBI treated tall goldenrod pulp (P-0) has peaks that are characteristic of both cellulose type I (cellulose I) and cellulose type II (cellulose II), with a broad amorphous peak and crystalline peaks at 2*θ* values of approximately 12.5°, 17°, 20°, and 22°. However, upon comparing the XRD spectra for P-50, P-100, P-200, and P-300, it can be seen that the diffraction peak intensities at 2*θ* values of approximately 12.5°, 20°, and 22° associated with the (1$\overset{¯}{1}$0), (110), and (020) planes, respectively, of the monoclinic unit cell of cellulose II, increase gradually with dose, while the diffraction peak intensities at 2*θ* values near 17° and 22°, characteristic of cellulose I, decrease. In particular, the P-300 spectrum is typical of cellulose II, with the peak at 2*θ* ≈ 17°, which is characteristic of cellulose I, entirely absent. Generally, cellulose II is derived from cellulose I via an alkali treatment or a solubilizing and recrystallization process [34,35,36]. The characteristic peaks for cellulose II of P-0 may be attributed to strong-alkali cooking by 12 wt.% NaOH solution and insufficient washing of the bleached pulp samples after bleaching treatment by an alkaline NaOCl solution. However, in this study, the loss of the main cellulose I peak with the use of a higher irradiation dose may be a result of the presence of the cellulose II in micro-sized cellulose fibers acting as a “seeds”, and thus inducing the rearrangement of the internal fiber structures. As another possibility, EB-irradiated cellulose fibers used in this study can be another form of cellulose, giving XRD features similar to cellulose II. Through XRD analysis alone, it is not possible to establish that these peaks correspond to cellulose II. In the future, more research is needed to establish the influence of EBI on the crystalline structures of cellulose.

### 3.3. Thermal Stability

The thermal stability of cellulose nanofibers is important for various succeeding applications. Figure 6 illustrates the thermal properties of P-0, P-50, P-100, P-200, and P-300. The first weight loss occurs for all samples between 30 and 110 °C and is associated with the evaporation of moisture from within the fibers. The second weight loss is observed in the range of 250−390 °C, and this results from the thermal degradation of the cellulose fibers. In Figure 6, it can be observed that the more finely separated cellulose fibers begin to thermally degrade at a lower temperature and have lower thermal stability even though this has not introduced the sulfate groups by sulfuric acid hydrolysis. The results could instead be explained by the temperature of cellulose decomposition shifting to lower values due to a reduction of cellulose I content in the more finely separated cellulose fibers with the higher EBI doses [37]. However, for the samples consisting of the more finely separated fibers produced by the greater EBI dose, the total amounts of char yield at around 800 °C are larger with respect to that of T-0. This result for the higher char yields may be due to the dehydration reaction at lower temperature [38].

### 3.4. Mechanical Performance

Figure 7 displays tensile strengths for paper samples prepared using only the fiber samples of P-0 and P-50, P-100, P-200, and P-300, respectively, by only thermocompression method without the use of a binder. In the figure it is apparent that there is an increase in tensile strength with the decrease in the fiber diameters. Specially, the tensile strength value of the P-0 paper, which was prepared with fibers over of 10 μm in diameter, was around 6.68 MPa, whereas the tensile strength values of the P-50, P-100, P-200, and P-300 papers with various fiber diameters, as governed by the differing EBI doses used in their preparation, increased, and are approximately 8.85, 15.78, 28.98, and 31.32 MPa, respectively. Thus, the P-300 paper with fibers of less than 200 nm in diameter had a 4.7-fold enhancement in tensile strength with respect to the P-0 paper which contained fibers of greater than 10 μm in diameter. The enhancement in tensile strength may be associated with incorporation of surface hydroxyl groups between the high-aspect-ratio cellulose nanofibers.

## 4. Conclusions

This study substantiates the possibility of isolating cellulose nanofibers from pulps of invasive tall goldenrod harvested in Korea by a convenient method without the use of strong acid or a high-pressure homogenizer, by using EBI. The separation degree of the obtained fibers prepared by EBI were evaluated by SEM, UV–vis, XRD, and TGA, as well as in terms of their mechanical properties. In the SEM images, we saw that the fibers with 10 μm in diameter became more finely separated with increasing EBI dose. In particular, we were able to observe that the fibers of the P-300 sample were separated into long cellulose nanofibers of approximately 160 nm in diameter. In addition, the paper samples prepared from the more finely separated fibers by the higher-dose EBI had improved UV−vis transmittance. XRD analysis revealed that EBI-treatment of cellulose fibers gradually decreased cellulose I content, converting it into a different type of cellulose, which has a crystalline structure with three planes, similar to cellulose II observed in the XRD spectra, due to a rearrangement of the internal structures in the fibers that occurs with increasing EBI dose. Via TGA, we saw that the more finely separated cellulose fibers had lower thermal stability, as a result of their reduced cellulose I content, and increasing amounts of char yield at around 800 °C due to the dehydration reaction at lower temperature. We were also able to measure the tensile strengths of the papers and observed that this quantity increased with a decrease in the fiber diameters; i.e., increasing the EBI dose resulted in increased tensile strength due to the reduced size of the component fibers.

## Figures and Tables

**Figure 1 nanomaterials-09-01358-f001:**
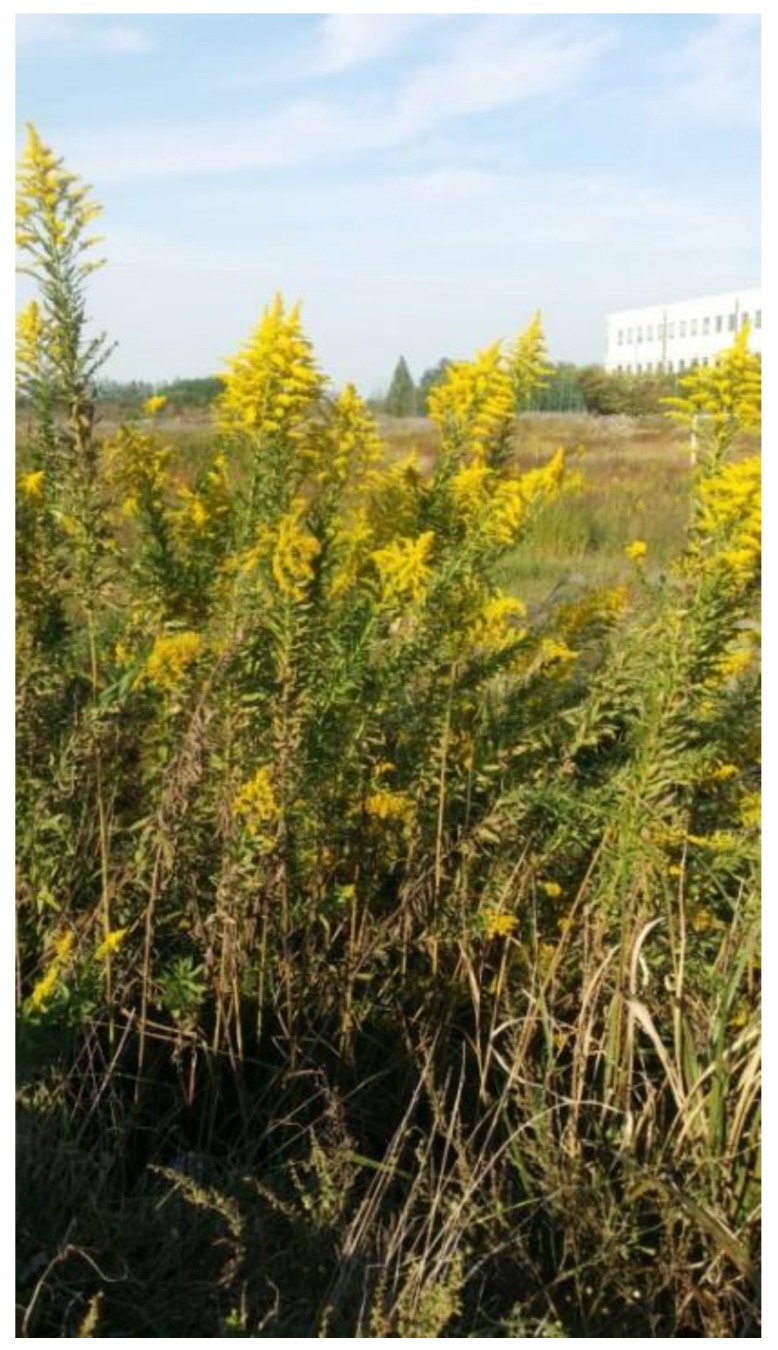
Photographs of invasive alien plant “tall goldenrod” (*Solidago altissima* L.) in Korea.

**Figure 2 nanomaterials-09-01358-f002:**
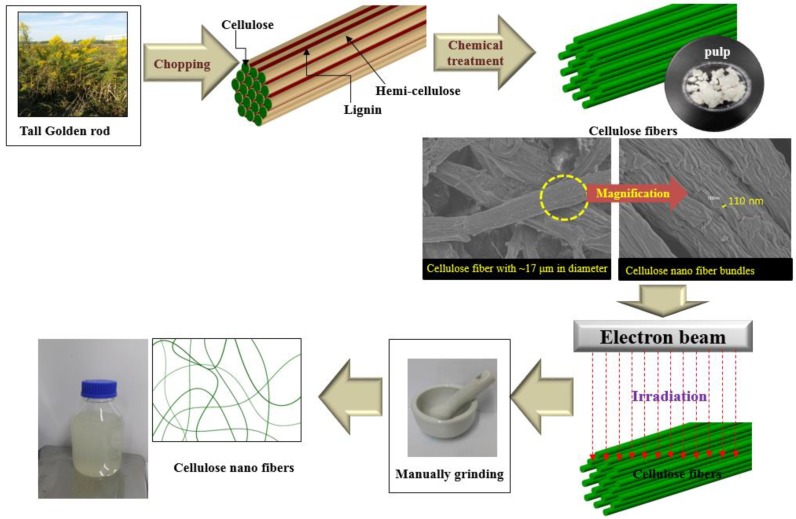
Schematic diagram of the preparation of cellulose nanofibers from tall goldenrod.

**Figure 3 nanomaterials-09-01358-f003:**
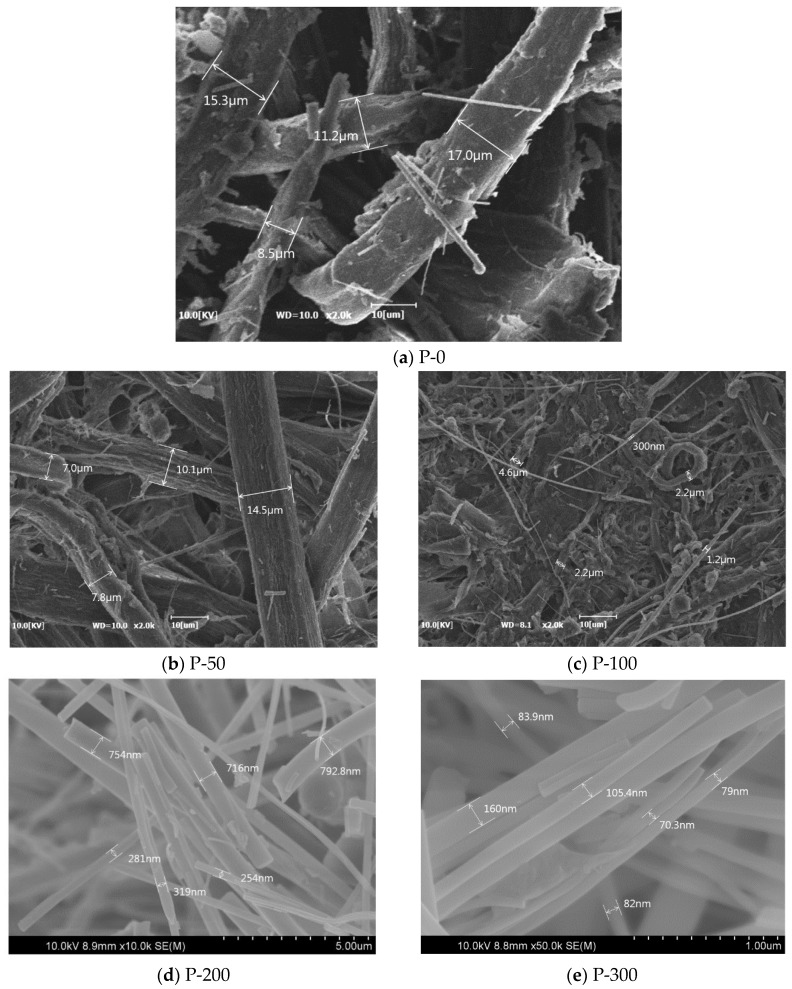
Scanning electron microscopy (SEM) images showing the degree of separation of (**a**) P-0, (**b**) P-50, (**c**) P-100, (**d**) P-200, and (**e**) P-300.

**Figure 4 nanomaterials-09-01358-f004:**
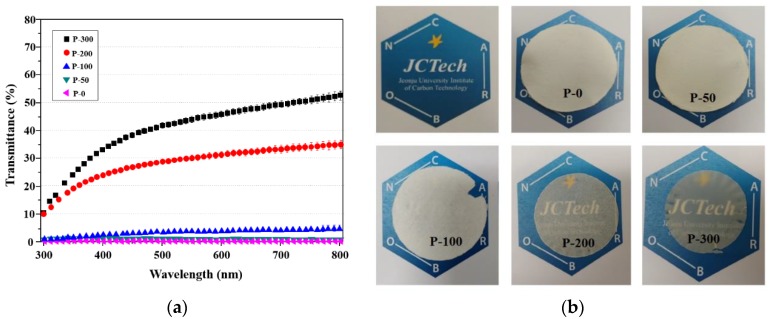
Visible light transmission performance. (**a**) UV–vis transmission spectra and (**b**) photographs of paper samples prepared from P-0, P-50, P-100, P-200, and P-300, respectively.

**Figure 5 nanomaterials-09-01358-f005:**
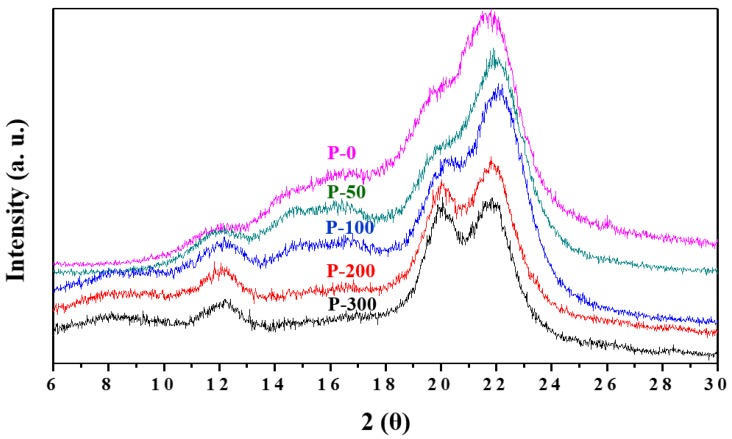
Comparison of XRD analysis of P-0, P-50, P-100, P-200, and P-300.

**Figure 6 nanomaterials-09-01358-f006:**
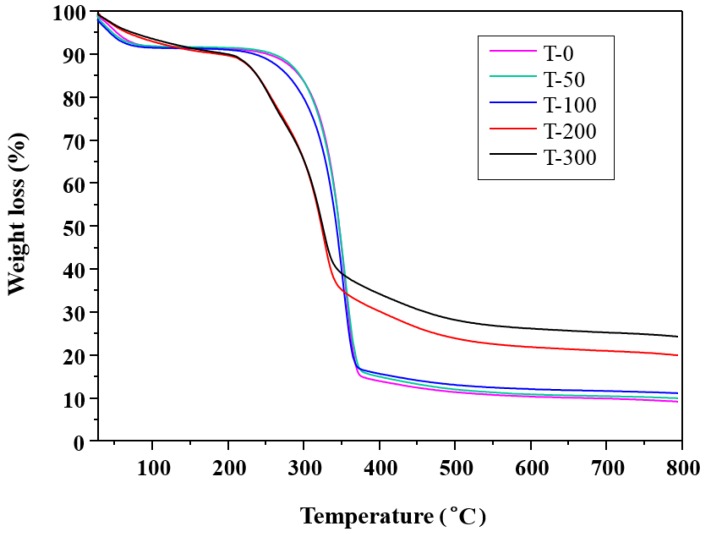
Thermogravimetric analysis results showing different thermal stability for P-0, P-50, P-100, P-200, and P-300.

**Figure 7 nanomaterials-09-01358-f007:**
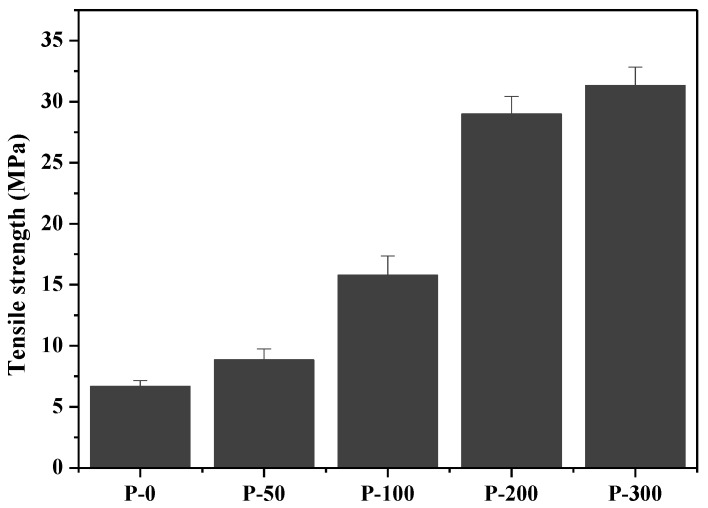
Tensile strengths for paper samples prepared using only the fiber samples of P-0, P-50, P-100, P-200, and P-300 fiber samples.

**Table 1 nanomaterials-09-01358-t001:** Alkali cooking and bleaching condition for the extraction of tall goldenrod cellulose fibers.

	Alkali Cooking	1st Bleaching	2nd Bleaching
Chemicals	12 wt.% sodium hydroxide	2 wt.% sodium chlorite 3 wt.% acetic acid	1.2 wt.% sodium hypochlorite
Temperature	120 °C	70 °C	Room temperature
Time	120 min	90 min	60 min

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
