# Peer review of "Electron Beam Irradiation Isolates Cellulose Nanofiber from Korea “Tall Goldenrod” Invasive Alien Plant Pulp"

_nanomaterials, 2019, doi:10.3390/nano9101358_

Round 1

Reviewer 1 Report

This is a fascinating document, with potential to be a very important article, attracting high interest. But the manner in which the results are presented and interpreted, in the present version, can be expected to reduce the impact and usefulness of the article.  It appears that the authors have stumbled onto a truly important finding, but they seem confused about what they have discovered.  Publishing the article in its present form would be a major mistake, in my opinion.

Detailed comments:

It is well known that cellulose I can be converted to cellulose II by exposure to conditions that either completely dissolve the cellulose or which substantially dissolved the cellulose, allowing the macromolecular chains to rearrange themselves in anti-parallel manner.  This cannot happen (usually) unless the process involves solubilization.  The fact that the XRD evidence suggests such a transformation might have three explanations, which need to be considered carefully by the authors:  (a)  partial transformation to cellulose II in the material prior to e-beam exposure suggests that the NaOH concentration in alkaline pulping was high relative to ordinary pulping, leading to an unusual material;  (b)  it might be hypothesized that the cellulose II present in the material might act as a “seed” for further rearrangement to the cellulose to cellulose II;  (c) it might be hypothesized that the effect of the e-beam exposure was mainly to break down the cellulose 1 crystal structure, converting a lot of it to amorphous cellulose or an intermediate structure with a lot of defects in it; (d) it might be hypothesized that the crystalline form was changed from cellulose I to another form of cellulose, but not cellulose II (since it is hard to imagine inversion of the chain directions under dry conditions). The words “eco-friendly method” on the second line of the abstract (and elsewhere in the article) cannot be justified, in my opinion. The approach used by the authors is very energy-intensive, which is a big “no-no” for environmental impact.  To make matters worse, the authors used very eco-unfriendly bleaching chemicals (sodium chlorite and especially sodium hypochlorite), which can be expected to general highly toxic chlorinated aromatic compounds. Lines 87-90: It is necessary to carefully define the condition of the specimens at the time of application of the EBI.  Was the material dried before exposure to the beam?  What were the conditions of drying?  What was the moisture content?  Was there any evidence of scorching, especially under the highest conditions of exposure? Line 122: Please arrange the text and figure so that the text introducing a figure precedes the figure, at least in part.  The SEM images do not “tell a story” without explanation. Line 132: The term “fiber bundles” is not correct. Those are “fibers” shown in the first SEM image. Line 169: Again, move the text introducing the XRD and FTIR figures to before the figures. Figure 5: The XRD results do not necessarily show an increase in cellulose II due to application of the electron beam.  Rather, the results are consistent with the selective degradation of cellulose I.  This explanation seems most likely, since it is not reasonable to expect switching of the direction of half of the cellulose chains under dry conditions, as would be required for a conversion from cellulose I to cellulose II.  The literature shows that such conversion requires full or at least partial solubilization (or temporary derivitization) of the cellulose. Even if the main finding of this work was SELECTIVE BREAKDOWN OF CELLULOSE I by the e-beam, that would still be a major breakthrough, and it would make the authors famous and bring credit to the journal. Figure 6: The figure is confusing because it does not appear to show any hydrogens.  It is very difficult to convey concepts of hydrogen bonding when using structures that do not include the hydrogens, especially those hydrogens attached to oxygens. Figure 7: The TGA results are consistent with the effect of the e-beam causing breakdown of or multiple defects in the crystal structure of the cellulose I. Line 211: The phrase “result of a larger surface area” does not appear to be a mechanistically sound explanation.  There is no physical reason to expect that “surface area” would play any role in the mechanism. Line 234: Again, the words “eco-friendly method” are not justified.  The use of a highly energy-intensive method is definitely NOT eco-friendly. Line 242: The conclusion that cellulose I was converted to cellulose II is based on inadequate evidence, is physically unlikely, and seems likely to expose the authors to ridicule unless they delete the phrase before publication. Line 245: Again, the words “due to a larger specific surface area” are not consistent with any reputable theory.  Correlation does not equal causation, especially if you have are comparing two items.

Author Response

Reviewer 1

Thank you for the reviewers for their very constructive comments on this manuscript.

Revision was done as follows.

It is well known that cellulose I can be converted to cellulose II by exposure to conditions that either completely dissolve the cellulose or which substantially dissolved the cellulose, allowing the macromolecular chains to rearrange themselves in anti-parallel manner. This cannot happen (usually) unless the process involves solubilization. The fact that the XRD evidence suggests such a transformation might have three explanations, which need to be considered carefully by the authors: (a) partial transformation to cellulose II in the material prior to e-beam exposure suggests that the NaOH concentration in alkaline pulping was high relative to ordinary pulping, leading to an unusual material; (b) it might be hypothesized that the cellulose II present in the material might act as a “seed” for further rearrangement to the cellulose to cellulose II; (c) it might be hypothesized that the effect of the e-beam exposure was mainly to break down the cellulose 1 crystal structure, converting a lot of it to amorphous cellulose or an intermediate structure with a lot of defects in it; (d) it might be hypothesized that the crystalline form was changed from cellulose I to another form of cellulose, but not cellulose II (since it is hard to imagine inversion of the chain directions under dry conditions).

→Your hypothesis and advice have helped me a lot. Based on your advice, I revised as follows.

Before revision:

The XRD spectra plotted in Figure 5 (a) show the crystalline structure variation between the P-0 and P-50, P-100, P-200, and P-300 pulp samples, obtained by treating tall goldenrod pulp with various EBI dose. As shown in Figure 5, non-EBI treated tall goldenrod pulp (P-0) has peaks that are characteristic of both cellulose type I (cellulose I) and cellulose type II (cellulose II), with a broad amorphous peak and crystalline peaks at 2θ values of approximately 12.5°, 17°, 20°, and 22°. However, upon comparing the XRD spectra for P-50, P-100, P-200, and P-300, it is clear that the diffraction peak intensities at 2θ values of approximately 12.5°, 20°, and 22° associated with the (1­­0), (102), and (200) planes, respectively, of the monoclinic unit cell of cellulose II, increase gradually with dose, while the diffraction peak intensities at 2θ values near 17° and 22°, characteristic of cellulose I, decrease. In particular, the P-300 spectrum is typical of cellulose II, with the peak at 2θ ≈ 17°, which is characteristic of cellulose I, entirely absent. Generally, cellulose II is derived from cellulose I via an alkali treatment or a solubilizing and recrystallization process [34-36]. The characteristic peaks for cellulose II of P-0 may be attributed to insufficient washing of the bleached pulp samples after bleaching treatment by an alkaline NaOCl solution. However, in this study, the loss of the main cellulose I peak with the use of a higher irradiation dose may be a result of the EBI permeating the crystalline organization of the micro-sized cellulose fibers and triggering the weakening and breaking of intermolecular and intramolecular bonds, thus inducing the rearrangement of the internal fiber structures. In the FT-IR spectra shown in Figure 5 (b), hydrogen-bonded O−H stretching and C−H stretching are respectively observed at 3400−3000 cm-1 and 2900−2800 cm-1 [37, 38]. However, we can see that the peak at around 3313 cm-1 in the spectrum of P-0, the non-EBI-treated cellulose fiber, is shifted to higher wave numbers, to approximately 3400 cm-1, when the EBI dose increase, because of the formation of the 2−OH···O−6 intramolecular hydrogen bond (indicated by a purple circle in Figure 6) with the conversion from cellulose I to cellulose II [39]. In addition, an increase in the intensity at 895 cm-1 can be assigned to the crystalline structure differences between cellulose I and cellulose II [40,41]. Therefore, combining the XRD results with the FT-IR analysis, it can be suggested that EBI provides the possibility to convert cellulose I into cellulose II without chemical treatment.

After revision:

The XRD spectra plotted in Figure 5 show the crystalline structure variation between the P-0 and P-50, P-100, P-200, and P-300 pulp samples, obtained by treating tall goldenrod pulp with various EBI dose. As shown in Figure 5, non-EBI treated tall goldenrod pulp (P-0) has peaks that are characteristic of both cellulose type I (cellulose I) and cellulose type II (cellulose II), with a broad amorphous peak and crystalline peaks at 2θ values of approximately 12.5°, 17°, 20°, and 22°. However, upon comparing the XRD spectra for P-50, P-100, P-200, and P-300, it can be seen that the diffraction peak intensities at 2θ values of approximately 12.5°, 20°, and 22° associated with the (1­­0), (110), and (020) planes, respectively, of the monoclinic unit cell of cellulose II, increase gradually with dose, while the diffraction peak intensities at 2θ values near 17° and 22°, characteristic of cellulose I, decrease. In particular, the P-300 spectrum is typical of cellulose II, with the peak at 2θ ≈ 17°, which is characteristic of cellulose I, entirely absent. Generally, cellulose II is derived from cellulose I via an alkali treatment or a solubilizing and recrystallization process [34-36]. The characteristic peaks for cellulose II of P-0 may be attributed to strong-alkali cooking by 12 wt% NaOH solution and insufficient washing of the bleached pulp samples after bleaching treatment by an alkaline NaOCl solution. However, in this study, the loss of the main cellulose I peak with the use of a higher irradiation dose may be a result of the presence of the cellulose II in micro-sized cellulose fibers acting as a “seeds”, and thus inducing the rearrangement of the internal fiber structures. As another possibility, EB-irradiated cellulose fibers used in this study can be another form of cellulose similar to, but not the same as, cellulose II . Through XRD analysis alone, it is not possible to establish that these peaks correspond to cellulose II. In the future, more research is needed to establish the influence of EBI on the crystalline structures of cellulose.

The words “eco-friendly method” on the second line of the abstract (and elsewhere in the article) cannot be justified, in my opinion. The approach used by the authors is very energy-intensive, which is a big “no-no” for environmental impact. To make matters worse, the authors used very eco-unfriendly bleaching chemicals (sodium chlorite and especially sodium hypochlorite), which can be expected to general highly toxic chlorinated aromatic compounds.

→ In this study, the words of “eco-friendly method” are about the process to nanocellulose fiber isolation from pulps. As your comments, pulping process from tall golden rod plants is not eco-friendly method. Bleaching process is only for attaching the reference. Another bleaching process using H2O2 is proper but I could not find as the reference for bleaching of pulp. However, as your comments, we deleted the word of “eco-friendly”.

Lines 87-90: It is necessary to carefully define the condition of the specimens at the time of application of the EBI. Was the material dried before exposure to the beam? What were the conditions of drying? What was the moisture content? Was there any evidence of scorching, especially under the highest conditions of exposure?

→ The obtained pulps are lyophilized and kept in a desiccator and there was no evidence of scorching because the pulps are irradiated at a scanned beam of 1.14 MeV accelerating voltage.

Line 122: Please arrange the text and figure so that the text introducing a figure precedes the figure, at least in part. The SEM images do not “tell a story” without explanation. Line 132: The term “fiber bundles” is not correct. Those are “fibers” shown in the first SEM image.

→ I revised as your comments and added the contents between line 125 and line 129.

I deleted the word of “bundles”

After revision:

This effect can be explained by the fact that the EBI treatment permeates weak parts between nanocellulose fibers within micro-sized cellulose fibers, inducing wakening and breaking. Then, by manually grinding the cellulose fibers using a mortar and pestle, the degree of separation of within the cellulose fibers is enhanced as the sections between nanocellulose fibers already weakened by EBI break. The higher EBI dose generated more finely cellulose fibers.

Line 169: Again, move the text introducing the XRD and FTIR figures to before the figures. Figure 5: The XRD results do not necessarily show an increase in cellulose II due to application of the electron beam. Rather, the results are consistent with the selective degradation of cellulose I. This explanation seems most likely, since it is not reasonable to expect switching of the direction of half of the cellulose chains under dry conditions, as would be required for a conversion from cellulose I to cellulose II. The literature shows that such conversion requires full or at least partial solubilization (or temporary derivitization) of the cellulose. Even if the main finding of this work was SELECTIVE BREAKDOWN OF CELLULOSE I by the e-beam, that would still be a major breakthrough, and it would make the authors famous and bring credit to the journal. Figure 6: The figure is confusing because it does not appear to show any hydrogens. It is very difficult to convey concepts of hydrogen bonding when using structures that do not include the hydrogens, especially those hydrogens attached to oxygens.

→ Based on your advice, I deleted Figure 6 because it could cause confusion.

Figure 7: The TGA results are consistent with the effect of the e-beam causing breakdown of or multiple defects in the crystal structure of the cellulose I. Line 211: The phrase “result of a larger surface area” does not appear to be a mechanistically sound explanation. There is no physical reason to expect that “surface area” would play any role in the mechanism.

→ I revised as your comment

Before: The effect may instead be the result of a larger surface area exposing the heat source as cellulose fibers are more finely separated by the higher EBI doses

After revision: The results could instead be explained by the temperature of cellulose decomposition shifting to lower values due to a reduction of cellulose I content in the more finely separated cellulose fibers with the higher EBI doses

Line 234: Again, the words “eco-friendly method” are not justified. The use of a highly energy-intensive method is definitely NOT eco-friendly.

I deleted the word “eco-friendly” as your commnent

Line 242: The conclusion that cellulose I was converted to cellulose II is based on inadequate evidence, is physically unlikely, and seems likely to expose the authors to ridicule unless they delete the phrase before publication.

→ I revised as following

Before revision: FT-IR and XRD analysis revealed that EBI-treatment of cellulose fibers gradually converted cellulose I into a different type of cellulose, cellulose II, which has crystalline structures with three planes observed in the XRD spectra, due to a rearrangement of the internal structures in the fibers that occurs with increasing EBI dose

After revision: XRD analysis revealed that EBI-treatment of cellulose fibers gradually decreased cellulose I content, converting it into a different type of cellulose, which has a crystalline structure with three planes, similar to cellulose II observed in the XRD spectra, due to a rearrangement of the internal structures in the fibers that occurs with increasing EBI dose.

Line 245: Again, the words “due to a larger specific surface area” are not consistent with any reputable theory. Correlation does not equal causation, especially if you have are comparing two items.

→ I revised as your comment.

Before revision: we saw that finely-separated cellulose fibers resulted in decreasing more the thermal stability due to larger specific surface area of fiber in contact with the heat source

After revision: the more finely separated cellulose fibers had lower thermal stability, as a result of their reduced cellulose I content,

Thank you again for your review of this paper

Best regards.

Hye Kyoung Shin

Reviewer 2 Report

This manuscript demonstrated a unique approach for preparation of cellulose nanofiber by electron beam irradiation (EBI).  Though the cellulose nanofiber is not a new topic, the present method is still interesting to me. I would like to recommend it for Nanomaterials if the following questions can be addressed after revision.

After the bleaching treatment of goldenrod pulp, did the authors wash and dry the pulp? if so, what is the washing and drying conditions? How much distilled water was added to wet the pulp and how to determine the end of grinding, or how to guarantee people can repeat your results? How much sample was used for the EBI treatment per batch? How to prepare the paper samples with ground fibers, what is the thickness of the resultant paper for each condition, and how many repetitions for each condition in transmittance and tensile strength tests? Can you provide an error bar for each condition? In line 174-175, the authors stated that the “2θ values of approximately 12.5°, 20°, and 22° associated with the (11̅0), (102), and (200) planes, respectively, of the monoclinic unit cell of cellulose II.” While in the line 177, the authors said that “while the diffraction peak intensities at 2θ values near 17° and 22°, characteristic of cellulose I, decrease.” So, the peak at 2θ ≈22° is from cellulose II or cellulose I ? Which confused me. While, based on my knowledge,(Let me know if I am wrong), the peak at 2θ ≈19°( or 20°) is from the amorphous background, and the peak at 2θ ≈22.6 was assigned to cellulose I, the peak at 2θ ≈21.7 is supposed to be from cellulose II. (please check reference@ Moon, R., Martini, A., Nairn, J., Simonsen, J., Youngblood, J. (2011) Cellulose nanomaterials review: structure, properties and nanocomposites. Chem. Soc. Rev. 40:3941–3994; and @ Modern methods of determining crystallinity in cellulose, Pure and Applied Chemistry, Volume 5, Issue 1-2, Pages 91–106). So, I would like to suggest the authors double check their references and reassign the peaks if necessary. The authors try to use the shift and intensity change of peaks at 3400 cm-1 and 895 cm-1 in FT-IR spectra to suppose the EBI can convert cellulose I into cellulose II, which is kind of inadequate to me, I would like to suggest the authors modified their text with a more conservative statement, or use the method in the following reference to give solid evidence. (Quantitative analysis of the transformation process of cellulose I - cellulose II using NIR FT Raman spectroscopy and chemometric methods, Cellulose (2009) 16:407–415). If possible, I would like to see the author give an explanation about the mechanism of this process, i.e. why the EBI can be used to fabricate cellulose nanofibers? The following mistakes should be corrected. Line 204, “Fig 6” should be “Figure 7” Line 154, should remove one “separation of “ in “the finer separation of separation of fibers with increasing EBI” . Line 220, “Figure 7” should be “Figure 8”

Author Response

Reviewer 2

Thank you for the reviewers for their very constructive comments on this manuscript.

Revision was done as follows.

After the bleaching treatment of goldenrod pulp, did the authors wash and dry the pulp? if so, what is the washing and drying conditions?

→ We washed over 10 times, and the washed pulps were lyophilized and kept in a desiccator.

How much distilled water was added to wet the pulp

→ Distilled water weight: the pulps weight (10:1)

How to determine the end of grinding,

→ I determined the end of grinding using time (during 1 min)

or how to guarantee people can repeat your results?

→ Anyone can easily get the same result as us if they follow the experimental method of our paper.

How much sample was used for the EBI treatment per batch?

→ EBI treatment were conducted using the scanned beam of conveyor type. Therefore, a considerable amount of pulps can be irradiated by EB.

How to prepare the paper samples with ground fibers,

→ The paper samples were prepared by filtering ground fibers (dry weight: 0.1 g) mixed with water using glass filter assembly instrument.

what is the thickness of the resultant paper for each condition,

→The thickness of paper samples is similar to general A4 paper.

and how many repetitions for each condition in transmittance and tensile strength tests? Can you provide an error bar for each condition?

→ As your comments, I revised the graphs of transmittance (repetition: 6 times) tensile strength (repetition: 10 times ) into graph using an error bar.

In line 174-175, the authors stated that the “2θ values of approximately 12.5°, 20°, and 22° associated with the (11̅0), (102), and (200) planes, respectively, of the monoclinic unit cell of cellulose II.” While in the line 177, the authors said that “while the diffraction peak intensities at 2θ values near 17° and 22°, characteristic of cellulose I, decrease.” So, the peak at 2θ ≈22° is from cellulose II or cellulose I ?

→ 2θ ≈22° is from cellulose II and 2θ ≈22.6° is from cellulose II.

Which confused me. While, based on my knowledge,(Let me know if I am wrong), the peak at 2θ ≈19°( or 20°) is from the amorphous background, and the peak at 2θ ≈22.6 was assigned to cellulose I, the peak at 2θ ≈21.7 is supposed to be from cellulose II. (please check reference@ Moon, R., Martini, A., Nairn, J., Simonsen, J., Youngblood, J. (2011) Cellulose nanomaterials review: structure, properties and nanocomposites. Chem. Soc. Rev. 40:3941–3994; and @ Modern methods of determining crystallinity in cellulose, Pure and Applied Chemistry, Volume 5, Issue 1-2, Pages 91–106). So, I would like to suggest the authors double check their references and reassign the peaks if necessary. The authors try to use the shift and intensity change of peaks at 3400 cm-1 and 895 cm-1 in FT-IR spectra to suppose the EBI can convert cellulose I into cellulose II, which is kind of inadequate to me, I would like to suggest the authors modified their text with a more conservative statement, or use the method in the following reference to give solid evidence. (Quantitative analysis of the transformation process of cellulose I - cellulose II using NIR FT Raman spectroscopy and chemometric methods, Cellulose (2009) 16:407–415). If possible, I would like to see the author give an explanation about the mechanism of this process, i.e. why the EBI can be used to fabricate cellulose nanofibers? The following mistakes should be corrected. Line 204, “Fig 6” should be “Figure 7” Line 154, should remove one “separation of “ in “the finer separation of separation of fibers with increasing EBI” . Line 220, “Figure 7” should be “Figure 8”

→ Based on you and another reviewers’ comments, I revised for XRD analysis and deleted Figure 6 and FT-IR analysis parts because it could cause confusion.

I revised as follows

Before revision:

The XRD spectra plotted in Figure 5 (a) show the crystalline structure variation between the P-0 and P-50, P-100, P-200, and P-300 pulp samples, obtained by treating tall goldenrod pulp with various EBI dose. As shown in Figure 5, non-EBI treated tall goldenrod pulp (P-0) has peaks that are characteristic of both cellulose type I (cellulose I) and cellulose type II (cellulose II), with a broad amorphous peak and crystalline peaks at 2θ values of approximately 12.5°, 17°, 20°, and 22°. However, upon comparing the XRD spectra for P-50, P-100, P-200, and P-300, it is clear that the diffraction peak intensities at 2θ values of approximately 12.5°, 20°, and 22° associated with the (1­­0), (102), and (200) planes, respectively, of the monoclinic unit cell of cellulose II, increase gradually with dose, while the diffraction peak intensities at 2θ values near 17° and 22°, characteristic of cellulose I, decrease. In particular, the P-300 spectrum is typical of cellulose II, with the peak at 2θ ≈ 17°, which is characteristic of cellulose I, entirely absent. Generally, cellulose II is derived from cellulose I via an alkali treatment or a solubilizing and recrystallization process [34-36]. The characteristic peaks for cellulose II of P-0 may be attributed to insufficient washing of the bleached pulp samples after bleaching treatment by an alkaline NaOCl solution. However, in this study, the loss of the main cellulose I peak with the use of a higher irradiation dose may be a result of the EBI permeating the crystalline organization of the micro-sized cellulose fibers and triggering the weakening and breaking of intermolecular and intramolecular bonds, thus inducing the rearrangement of the internal fiber structures. In the FT-IR spectra shown in Figure 5 (b), hydrogen-bonded O−H stretching and C−H stretching are respectively observed at 3400−3000 cm-1 and 2900−2800 cm-1 [37, 38]. However, we can see that the peak at around 3313 cm-1 in the spectrum of P-0, the non-EBI-treated cellulose fiber, is shifted to higher wave numbers, to approximately 3400 cm-1, when the EBI dose increase, because of the formation of the 2−OH···O−6 intramolecular hydrogen bond (indicated by a purple circle in Figure 6) with the conversion from cellulose I to cellulose II [39]. In addition, an increase in the intensity at 895 cm-1 can be assigned to the crystalline structure differences between cellulose I and cellulose II [40,41]. Therefore, combining the XRD results with the FT-IR analysis, it can be suggested that EBI provides the possibility to convert cellulose I into cellulose II without chemical treatment.

After revision:

The XRD spectra plotted in Figure 5 show the crystalline structure variation between the P-0 and P-50, P-100, P-200, and P-300 pulp samples, obtained by treating tall goldenrod pulp with various EBI dose. As shown in Figure 5, non-EBI treated tall goldenrod pulp (P-0) has peaks that are characteristic of both cellulose type I (cellulose I) and cellulose type II (cellulose II), with a broad amorphous peak and crystalline peaks at 2θ values of approximately 12.5°, 17°, 20°, and 22°. However, upon comparing the XRD spectra for P-50, P-100, P-200, and P-300, it can be seen that the diffraction peak intensities at 2θ values of approximately 12.5°, 20°, and 22° associated with the (1­­0), (110), and (020) planes, respectively, of the monoclinic unit cell of cellulose II, increase gradually with dose, while the diffraction peak intensities at 2θ values near 17° and 22°, characteristic of cellulose I, decrease. In particular, the P-300 spectrum is typical of cellulose II, with the peak at 2θ ≈ 17°, which is characteristic of cellulose I, entirely absent. Generally, cellulose II is derived from cellulose I via an alkali treatment or a solubilizing and recrystallization process [34-36]. The characteristic peaks for cellulose II of P-0 may be attributed to strong-alkali cooking by 12 wt% NaOH solution and insufficient washing of the bleached pulp samples after bleaching treatment by an alkaline NaOCl solution. However, in this study, the loss of the main cellulose I peak with the use of a higher irradiation dose may be a result of the presence of the cellulose II in micro-sized cellulose fibers acting as a “seeds”, and thus inducing the rearrangement of the internal fiber structures. As another possibility, EB-irradiated cellulose fibers used in this study can be another form of cellulose similar to, but not the same as, cellulose II . Through XRD analysis alone, it is not possible to establish that these peaks correspond to cellulose II. In the future, more research is needed to establish the influence of EBI on the crystalline structures of cellulose.

Thank you again for your review of this paper

Best regards.

Hye Kyoung Shin

Reviewer 3 Report

The manuscript can be rejected. The authors use totally confusing and not existing terminology to describe the subject of the research. "nano-cellulose fibers " this does NOT exist! Maybe it is a fancy word, but not scientifically correct. I am aware that many people use that and can be found high impact journal articles, but this is a totally confusing term and we should teach the next generation, it our responsibility what we give them, it is very important to decribe correctly the meaning of our research anytime. 

Author Response

Reviewer 3

Thank you for the reviewers for their very constructive comments on this manuscript.

The manuscript can be rejected. The authors use totally confusing and not existing terminology to describe the subject of the research. "nano-cellulose fibers " this does NOT exist! Maybe it is a fancy word, but not scientifically correct. I am aware that many people use that and can be found high impact journal articles, but this is a totally confusing term and we should teach the next generation, it our responsibility what we give them, it is very important to decribe correctly the meaning of our research anytime. 

→ We obtained nanocellulose fibers from pulps by a convenient method via EBI treatment as shown below photos.

We didn't just happen to find it, but it's the result of a reproducibility experiment that we've been working on for years. And, as your comments, for teaching the next generation, we've revised a lot of contents of manuscript based on the advice of another reviewer.

Thank you again for your review of this paper

Best regards.

Hye Kyoung Shin

Round 2

Reviewer 1 Report

The authors did a careful job in incorporating the input from the reviewers.

On line 188 I want to suggest a change in wording.  I would recommend replacing the text "similar to, but not the same as" to "giving XRD features similar to".

Otherwise, the article seems ready for publication.

Author Response

Reviewer 1

Thank you for the reviewers for their very constructive comments on this manuscript.

Revision was done as follows.

On line 188 I want to suggest a change in wording.  I would recommend replacing the text "similar to, but not the same as" to "giving XRD features similar to".

→ I revised "similar to, but not the same as" into "giving XRD features similar to" as your comment

Thank you again for your review of this paper

Best regards.

Hye Kyoung Shin

Reviewer 2 Report

In the revised manuscript, there is still lack of detailed description on the experimental part, like how many times to wash the pulp; how long to grind the fibers; how much fiber was used to prepare the final papers and what is the exact thickness of the papers. Without the accurate numbers, it is not possible to let others to repeat the present results (i.e. the transmittance and tensile strength could be different with varied paper thickness.).

The numbers of repetitions should be mentioned in the main text, not just tell the reviewers.

The  FTIR result was removed from the result part, so it is no necessary keep it in the Analysis part (2.3).

The latest discussion about the XRD spectra is still not convinced to me, (like the the characteristic peaks for cellulose II in P0 sample, etc.), while, I believed the authors already tried their best, so I don't think they can further improve it. The readers have to accept as is.

There is still lack of an convinced explanation about the mechanism of the presnet process, while, it is not the main purpose of this paper, so I am fine with the current explanation.

Author Response

Reviewer 2

Thank you for the reviewers for their very constructive comments on this manuscript.

Revision was done as follows.

In the revised manuscript, there is still lack of detailed description on the experimental part, like how many times to wash the pulp; how long to grind the fibers; how much fiber was used to prepare the final papers and what is the exact thickness of the papers. Without the accurate numbers, it is not possible to let others to repeat the present results (i.e. the transmittance and tensile strength could be different with varied paper thickness.).

→ As your comment, I added as follows:

After:

Line 80: The bleached pulps were lyophilized after washed over 10 times in water and then kept in a desiccator.

Line 92-93: EBI-treated cellulose fibers (0.2 g) were wetted with distilled water (2 g) and manually ground using a mortar and pestle for 1 minute.

Line 106: Ultraviolet-visible (UV-vis) light transmittance was performed on paper samples (the thickness: 0.14 ± 0.023 mm)

The  FTIR result was removed from the result part, so it is no necessary keep it in the Analysis part (2.3).

→ I deleted the FT-IR in the analysis part.

Thank you again for your review of this paper

Best regards.

Hye Kyoung Shin

Reviewer 3 Report

The manuscript still contain confusingly written part. What i can see from the microscopy images that they used cellulose fibers (lumen inside, have fiber-wall, elongated shape etc.) and that cellulose fibers have been treated with EBI than decompose it to cellulose micro- and nanofibrils, which is a fiber wall component. I could not observe any nano-fibre. Fiber is a nature created something, nano-fibrils has no lumen inside. If i would like to understand your way of thinking, you use the same term "fiber" to describe 2 totally different entity. Why? I still can not accept that wrong terminology.  

Author Response

Reviewer 3

Thank you for the reviewers for their very constructive comments on this manuscript.

The manuscript still contain confusingly written part. What i can see from the microscopy images that they used cellulose fibers (lumen inside, have fiber-wall, elongated shape etc.) and that cellulose fibers have been treated with EBI than decompose it to cellulose micro- and nanofibrils, which is a fiber wall component. I could not observe any nano-fibre. Fiber is a nature created something, nano-fibrils has no lumen inside. If i would like to understand your way of thinking, you use the same term "fiber" to describe 2 totally different entity. Why? I still can not accept that wrong terminology.  

→I know that “Fiber” is a natural or synthetic linear substance that is significantly longer than it is wide. So, in this study, yarn-like materials with nanoscale diameter obtained via EBI were called “fiber”. As your comments, they can be 2 totally different entity but we think that their shapes are fibers. In addition, cellulose nano fibers obtained via EBI in this study has no lumen inside.

Thank you again for your review of this paper

Best regards.

Hye Kyoung Shin
